# Higher Serum Testosterone Levels Associated with Favorable Prognosis in Enzalutamide- and Abiraterone-Treated Castration-Resistant Prostate Cancer

**DOI:** 10.3390/jcm8040489

**Published:** 2019-04-11

**Authors:** Shinichi Sakamoto, Maihulan Maimaiti, Minhui Xu, Shuhei Kamada, Yasutaka Yamada, Hiroki Kitoh, Hiroaki Matsumoto, Nobuyoshi Takeuchi, Kosuke Higuchi, Haruhito A. Uchida, Akira Komiya, Maki Nagata, Hiroomi Nakatsu, Hideyasu Matsuyama, Koichiro Akakura, Tomohiko Ichikawa

**Affiliations:** 1Department of Urology, Chiba University Hospital, Chiba 260-8670, Japan; marghulanmaimaiti@gmail.com (M.M.); xuminhui198666@yahoo.co.jp (M.X.); nob.takeuchi1014@gmail.com (N.T.); akirakomiya@mac.com (A.K.); ichikawa@vmail.plala.or.jp (T.I.); 2Department of Urology, Yokohama Rosai Hospital, Yokohama 222-0036, Japan; shu.ukmd.d@gmail.com (S.K.); makinagata1109@gmail.com (M.N.); 3Department of Urology, Asahi Central Hospital, Aashi 289-2511, Japan; yasutaka1205@olive.plala.or.jp (Y.Y.); nakatsu@hospital.asahi.chiba.jp (H.N.); 4Department of Urology, Japan Community Healthcare Organization Tokyo Shinjuku Medical Center, Shinjyuku 162-8543, Japan; hirokitoh@gmail.com (H.K.); akakurak@ae.auone-net.jp (K.A.); 5Department of Urology, Graduate School of Medicine, Yamaguchi University, Ube 755-0046, Japan; hmatsumo@yamaguchi-u.ac.jp (H.M.); hidde@yamaguchi-u.ac.jp (H.M.); 6Department of Urology, Funabashi Medical Center, Funabashi 273-8588, Japan; k_h1069k@yahoo.co.jp; 7Department of Chronic Kidney Disease and Cardiovascular Disease, Okayama University Graduate School of Medicine, Dentistry and Pharmaceutical Sciences, Okayama 700-0914, Japan; hauchida@okayama-u.ac.jp

**Keywords:** abiraterone, enzalutamide, prostate cancer, androgen deprivation therapy, testosterone, castration resistant prostate cancer

## Abstract

Testosterone plays a significant role in maintaining the tumor microenvironment. The role of the target serum testosterone (TST) level in enzalutamide- (Enza) and abiraterone (Abi)-treated castration-resistant prostate cancer (CRPC) patients was studied. In total, 107 patients treated with Enza and/or Abi at Chiba University Hospital and affiliated hospitals were studied. The relationships between progression-free survival (PFS), overall survival (OS), and clinical factors were studied by Cox proportional hazard and Kaplan–Meier models. In the Abi and Enza groups overall, TST ≥ 13 ng/dL (median) (Hazard Ratio (HR) 0.43, *p* = 0.0032) remained an independent prognostic factor for PFS. In the Enza group, TST ≥ 13 ng/dL (median) was found to be a significant prognostic factor (HR 0.28, *p* = 0.0044), while, in the Abi group, TST ≥ 12 ng/dL (median) was not significant (HR 0.40, *p* = 0.0891). TST showed significant correlation with PFS periods (*r =* 0. 32, *p* = 0.0067), whereas, for OS, TST ≥ 13 ng/dL (median) showed no significant difference in the Abi and Enza groups overall. According to Kaplan–Meier analysis, a longer PFS at first-line therapy showed a favorable prognosis in the Enza group (*p* = 0.0429), while no difference was observed in the Abi group (*p* = 0.6051). The TST level and PFS of first-line therapy may be considered when determining the treatment strategy for CRPC patients.

## 1. Introduction

Prostate cancer is one of the most commonly diagnosed cancers in men [1]. Since the historical discovery of Dr. Huggins, androgen deprivation therapy (ADT) has been the mainstay of the therapy for locally advanced or metastatic prostate cancer [2]. According to the current guidelines, the target serum testosterone (TST) level during androgen deprivation therapy for prostate cancer was defined as <50 ng/dL [3]. However, we have recently reported the clinical significance of serum TST levels <20 ng/dL in prostate cancer patients who received combined androgen blockade (CAB) therapy in Japanese patients [4]. Despite an early response to ADT, the majority of patients with advanced disease progress and become refractory to ADT because of the emergence of castration-resistant prostate cancer (CRPC) cells. Although a number of mechanisms have been proposed, the androgen receptor (AR) plays a central role in the development of CRPC [5,6,7]. Evidence also indicate that estrogen receptor (ER) drives prostate growth [8,9]. Since the AR and ER axes play a major role in the development of CRPC, it is a classical treatment strategy to block either of the pathways. However, after a certain interval, the tumor relapses by acquiring treatment resistance. Thus, the establishment of the optimal treatment sequence in CRPC is the primary concern. 

Some predictors were reported to be related to the response to enzalutamide (Enza) and abiraterone (Abi), such as the presence of AR splicing variants, early prostate-specific antigen (PSA) response, neutrophil-to-lymphocyte ratio (NLR), the presence of visceral metastases, and so on [10]. However, no definitive guideline to help determine which of the two drugs should be used has yet been established. Furthermore, a significant survival benefit was clinically identified in patients with high-volume castration-sensitive prostate cancer (CSPC) treated with ADT in combination with docetaxel in the CHAARTED and STAMPEDE trials [11,12,13,14], while the LATTITUDE and STAMPEDE trials indicated a significant survival advantage in patients with high-volume CSPC treated with ADT plus abiraterone [15,16]. The sequences of AR-targeted drugs and chemotherapeutic agents remain controversial. Therefore, it is of primary importance to establish useful prognostic factors to guide treatment strategies for individual CRPC patients. 

Although a series of studies indicated the clinical significance of TST level and response to ADT, limited evidence exists related to TST and response to novel AR-targeted drugs. Classically, low TST related to favorable prognosis in patients received vintage ADT [3,4]. However, a recent study indicated a clinical advantage of high TST in patients who received Abi [17]. Furthermore, the prognostic significance of TST in patients who received Enza remains to be investigated. 

Here, we studied the association between TST level and response to novel AR-targeted drugs. The present findings may thus help to determine the optimal treatment strategies for CRPC patients. 

## 2. Materials and Methods

### 2.1. Patient Selection and Clinical Variables

A total of 107 patients treated with Enza and/or Abi for prostate cancer at Chiba University Hospital and affiliated hospitals between 2014 and 2017 were retrospectively analyzed.

The TST was defined as the total TST. The prognostic values of the were level and other clinical factors were evaluated in association with PSA levels and progression-free survival (PFS). Patients treated with radiation as first-line therapy or radical prostatectomy, having a history of radiation to the pelvis, systemic chemotherapy, and use of 5 alpha-reductase inhibitors was not included. 

Age, body mass index (BMI), first-line PFS (PFS of patients treated with first-line ADT with LH-RH analogue/antagonist and bicalutamide), site of metastasis, Gleason score, PSA at baseline, TST, nadir TST, and time-to-nadir TST were included as clinical factors. The Architect Testosterone II^®^ device (Abbot Diagnostics, Lake Forest, IL, USA) was used to determine TST levels. 

### 2.2. Definition of PSA Progression

PSA failure was defined according to the definition of The Prostate Cancer Clinical Trials Working Group 2 (PCWG2): a rising PSA, >2 ng/mL higher than the nadir; the rise has to be at least 25% over the nadir and has to be confirmed by a second PSA determination at least three weeks later. Also, the patient is required to have castrated levels of testosterone (<50 ng/dL).

### 2.3. Definition of High-Volume Tumor

A high-volume was defined as the presence of visceral metastases or ≥4 bone lesions with ≥1 beyond the vertebral bodies and pelvis, on the basis of a previous report [11]. 

### 2.4. Institutional Approval

This study was approved by the Institutional Review Board of Chiba University Hospital (approval number 2279). 

### 2.5. Statistical Analysis

Univariate and multivariate Cox proportional models and the Kaplan–Meier method were used for statistical analyses. Factors with *p* < 0.05 in univariate analysis were included in multivariate analysis when assessing Cox proportional models. Welch’s *t*-test, Fisher test, and Wilcoxon’s signed rank test were used to assess the associations of TST and other clinical variables. Statistical computations were carried out using the JMP 13.0.0 software program (SAS Institute, Cary, NC, USA). Significance was set at *p* < 0.05. 

## 3. Results

The study population included 107 patients, of whom 89 were treated with Enza, and 46 were treated with Abi. Twenty-eight patients received sequential therapy with the novel AR-targeted drugs. The median follow-up time was 68.3 months from first-line ADT. The patients’ characteristics are shown in Table 1. The patients’ median age was 73.0 years. The median PSA levels were 30.1 ng/mL for Enza and 41.1 ng/dL for Abi. The rates of lymph node, lung, liver, and bone metastases were 31.78%, 8.41%, 5.61%, and 83.18%, respectively. The rates of previous steroid use and estramustine use in Enza and Abi were 40.19% and 46.73%, respectively. The median TST at the initiation of Enza was 13 ng/dL, and that of Abi was 12 ng/dL. Further information is shown in Table 1. 

Table 2 shows the results of the univariate and multivariate analyses of the prognostic factors of PFS in Enza and Abi overall. By univariate analysis, first-line PFS ≥ 15.4 months (Hazard Ratio (HR) 0.60, *p* = 0.0309), previous docetaxel (HR 2.44, *p* = <.0001), high volume (HR 1.73, *p* = 0.0198), C-reactive protein (CRP) ≥ 0.15 mg/dL (HR 1.87, *p* = 0.0173), extent of disease (EOD) score (HR 2.79, *p* = 0.0004), PSA ≥ 34.1 ng/mL (HR 1.73, *p* = 0.0143), TST ≥ 13 ng/dL (HR 0.26, *p* = <.0001), steroid use (HR 2.24, *p* = 0.0003), and estramustine use (HR 1.69, *p* = 0.0179) were found to be significant prognostic factors. By multivariate analysis, only TST ≥ 13 ng/dL (HR 0.31, *p* = 0.0365) remained as an independent prognostic factor. 

The prognostic value of TST was also confirmed in each Enza and Abi group.

As shown in Appendix A, TST ≥ 13 ng/dL (median) was a significant factor in univariate (HR 0.28, *p* = 0.0044) and multivariate analysis (HR0.08, *p* = 0.0032) in Enza-treated patients. However, as shown in Appendix A, TST ≥ 12 ng/dL (median) was not significant in univariate analysis (HR0.40, *p* = 0.0891) in Abi-treated patients.

Furthermore, the effect of te TST on the response to Enza and Abi was studied by the Kaplan–Meier model. As shown in Figure 1, higher TST levels (≥13 ng/dL) were related to significantly better PFS in Enza and Abi groups overall (*p* < 0.0001) and in Enza-treated patients (*p* = 0.0032) (Figure 1a,b), while higher TST levels (≥12 ng/dL) showed no prognostic difference in Abi-treated patients (*p* = 0.0881) (Figure 1c).

When the PFS periods were stratified by the TST level (8 to 20 ng/dL), higher TST levels were related to longer PFS in Enza and Abi groups, with a correlation coefficient of 0.32 (*p* = 0.0067) (Figure 2a,b). 

To study the patients’ characteristics, two groups (TST <13 ng/dL and TST ≥ 13 ng/dL) were compared with respect to various clinical factors. A level of TST ≥ 13 ng/dL was related to higher alkaline phosphatase (ALP) levels (*p* = 0.0158) (Table 3). 

Table 4 shows the results of the univariate and multivariate analyses of the prognostic factors of overall survival (OS) in Enza and Abi groups overall. By multivariate analysis, previous docetaxel (HR 3.04, *p* = 0.0038), visceral mets (HR 7.88, *p* = 0.0139), ALP (HR 2.51, *p* = 0.0090) and lactate dehydrogenase (LDH) (HR 2.95, *p* = 0.0033) remained as independent prognostic factors. On the other hand, TST was not a significant factor neither in univariate analysis (HR 0.99, *p* = 0.8750) (Table 4), nor in Kaplan–Meier analysis (*p* = 0.8748) (Figure 1d). When patients who received initial AR-targeted therapy were selected, TST (HR 0.39, *p* = 0.0086) and CRP (HR 2.03, *p* = 0.0487) remained significant predictive factor for PFS in multivariate analysis (Appendix A), while, for OS, no factor remained significant in multivariate analysis (Appendix A). 

Next, the effects of the first-line therapy on the responses to Enza and Abi were evaluated. As shown in Figure 3a, first-line PFS (median ≥ 15.4 months) was related to a favorable response (*p* = 0.0273) for Enza and Abi overall (Figure 3a). However, the first-line PFS did not affect PFS in Abi-treated patients (median ≥ 12.6 months) (*p* = 0.6105), while they significantly affected PFS in Enza-treated patients (median ≥ 17.9 months) (*p* = 0.0429) (Figure 3b,c). Since the median first-line PFS in the Abi group was around 12 months, the patients were divided on the basis of the first-line PFS of 12.0 months for Enza. It was evident that a long first-line PFS (≥12.0 months) was related to significantly longer PFS in the Enza group (*p* = 0.0046) (Figure 3d). 

The results of the sequential use of Enza and Abi were also examined. Although it was not significant, the subsequent usage of novel AR-targeted drugs reduced the PSA response rate in patients treated with both drug (Appendix A). Regarding PFS, a reduction in the response period was more evident in patients receiving Abi after Enza (first Abi 15.7 weeks vs. Enza before Abi 9.93 weeks) than in patients administered Enza after Abi (1st Enza 12 weeks vs Abi before Enza 11.1 weeks).

## 4. Discussion

The current data indicate that a higher TST level at the stage of CRPC predicted a favorable response to Enza. When patients treated with Enza and Abi were combined, serum TST levels remained as independent prognostic factors for PFS. Moreover, the characteristics of the prognostic factors showed distinct differences between Enza and Abi. The response to Enza was more closely related to the response to first-line therapy, while the response to Abi was rather more independent from the response to first-line therapy. On the basis of the current data, TST level and the PFS of first-line therapy may be considered when choosing the treatment strategy for CRPC patients.

The reason why a higher TST level was associated with a favorable response to Enza and Abi may be related to the AR dependency of prostate cancer. When comparing clinical factors between higher and lower TST groups, higher TST was related to a relatively higher initial PSA (low TST 58.57 ng/mL vs high TST 81.65 ng/mL), although it was not significant. Since the expression of PSA is mediated by the transcriptional activity of nuclear AR, a higher initial PSA value may represent a higher basal AR activity inside the tumor microenvironment [18]. As both Enza and Abi work through AR, the higher AR dependency of the tumor may predict a higher response to AR-targeted drugs. 

Another reason may be that a TST level of 13 ng/dL itself represents the remaining potential to target the AR-related pathway. The clinical significance of lowering the TST level below 50 ng/dL has been described in several reports [4,19]. Our group and others have previously reported that patients who achieved nadir TST < 20 ng/dL survived longer than those who did not [4,20,21,22]. Klotz et al. reported that patients with first-year nadir testosterone consistently >20 ng/dL had significantly higher risks of dying of prostate cancer [21]. These data indicated the clinical significance of lowering TST to <20 ng/dL in the tumor microenvironment [4]. However, these data were obtained when Enza and Abi were not on the market or were had a very limited use. 

On the contrary, our data indicate that higher serum TST > 13 ng/dL represented the longer PFS for Enza and Abi groups, which is a novelty. The current cut-off value of serum TST of 13 ng/dL may represent the remaining AR dependency of the tumor that can only be blocked by treating with novel AR-targeted therapy. If tumor relapse occurs with a TST level >13 ng/dL, then this tumor may contain more AR-dependent cells, compared to tumors with a relapse occurring with TST < 13 ng/dL. Thus, even with a TST of 20 ng/dL, intensive blockade of the AR pathway through novel AR-targeted drugs would be effective, especially among patients with a higher TST level. 

The clinical advantage of high TST was also reported by Ryan et al. [17]. Although the TST cut-off value was even lower (>8.6 ng/dL) compared to that in the present study, these authors found a higher TST to be related to a better response to Abi. On the other hand, in our study, a clinical advantage of high TST was not found for Abi-treated patients but was found for Enza-treated patients. 

Interestingly enough, the higher TST group (TST ≥ 13 ng/dL) showed a relatively higher ALP, initial PSA, and PSA at the start of treatment with a similar rate of the high-volume tumor. The rate of visceral metastases was relatively higher in the TST ≥ 13 ng/dL group. Since the rate of visceral metastases was low (10–26%), it will be necessary to objectively assess the clinical significance of visceral metastases in a large number of patients. However, the present data may indicate that the response to novel AR-targeted drugs may not be related to a high or low tumor burden but may rather depend on the AR dependency of the tumor. 

The current data indicate that the first-line PFS (12 months) predicted a favorable response to Enza (*p* = 0.0046), while the first-line PFS showed no association with the response to Abi (*p* = 0.6051). These data are in line with the findings of a previous report. Loriot et al. reported that the first-line ADT period predicted the response to novel AR-targeted drugs, mainly Enza [23]. Bellmunt et al. reported that the response to Abi was not related to the first-line ADT period [24]. The reason for this difference is not clear. However, one of the reasons may be that Enza shares a common mechanism with bicalutamide as an AR antagonist. Enza blocks AR with over a 30-fold higher affinity compared to bicalutamide [25]. The majority of the patients received CAB with bicalutamide and LH-RH agonist/antagonist as first-line ADT. Therefore, the patients who responded well to bicalutamide may also represent the patients who respond well to a potent AR antagonist, namely Enza. On the other hand, Abi works through Cyp17 inhibition, so the mechanism is distinct from that of bicalutamide and Enza. This mechanistic difference between Abi and Enza may represent the difference in response to the first-line therapy between Enza and Abi. 

The present study provides several important findings. (1) Higher TST (≥13 ng/dL) at the initiation of drug administration was associated with a favorable response to novel AR-targeted drugs, especially Enza; (2) The response to Enza was affected by the PFS of the first-line ADT, while the response to Abi was not affected by the PFS of the first-line ADT. 

There are several limitations associated with this study. First, the sample size was rather small, which limits the reliability of the analysis, especially when assessing Abi and Enza independently. Second, because of the limited follow-up periods and limited outcomes regarding OS, the assessment of prognostic factors was mainly based on PFS. Third, because of the limited number of patients, the analysis was not divided in pre- or post-chemo settings. Although TST levels remain as predictors even among the factors that included the previous usage of chemotherapy, the response rate may also be affected by the previous usage of chemotherapy. Fourth, because of the prior introduction of Enza in the Japanese healthcare system, the majority of patients in the Enza group (92%) received the drug as a first-line novel AR-targeted drug, while half of the patients (54%) received Abi as a first-line AR-targeted drug. The different results observed for Enza and Abi, including association with TST levels, may possibly be affected by the background of the patients who received the drugs. We are currently performing a prospective study to re-assess the role of TST in the initial usage of Enza and Abi. The effects of the differences in the clinical features of the patients that induced the treatment choice in the current manuscript will be answered in the near future. 

## 5. Conclusions

A higher TST level (≥13 ng/dL) was associated with a favorable prognosis in Enza- and/or Abi-treated patients. A TST level of 13 ng/dL may predict a favorable response to novel AR-targeted drugs in CRPC patients.

## Figures and Tables

**Figure 1 jcm-08-00489-f001:**
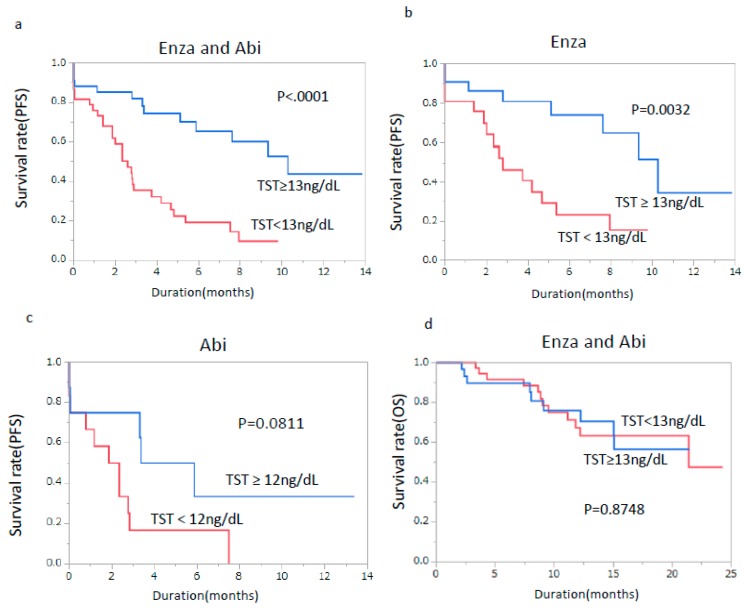
Progression-free survival according to TST levels. (**a**) Progression-free survival according to TST level <13 ng/dL or ≥13 ng/dL as the cut-off values in enzalutamide- and abiraterone-treated patients. (**b**) Progression-free survival according to TST level <13 ng/dL or ≥13 ng/dL as the cut-off values in enzalutamide-treated patients. (**c**) Progression-free survival according to TST level <12 ng/dL or ≥12 ng/dL as the cut-off values in abiraterone-treated patients. (**d**) Overall survival according to TST level <13 ng/dL or ≥13 ng/dL as the cut-off values in enzalutamide- and abiraterone-treated patients.

**Figure 2 jcm-08-00489-f002:**
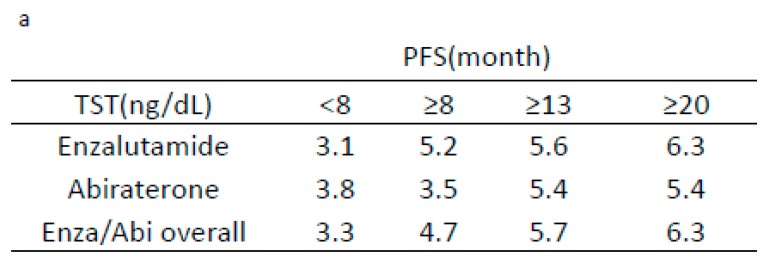
Progression-free survival based on the serum TST cut off level. (**a**) Progression-free survival according to TST level <8 ng/dL, ≥8 ng/dL, ≥13 ng/dL, or ≥20 ng/dL as the cut-off values in enzalutamide- and abiraterone-treated patients. (**b**) Correlation of serum TST level and progression-free survival in enzalutamide- and abiraterone-treated patients. The red circle indicates 95% of data used for the correlation analysis, to avoid errors.

**Figure 3 jcm-08-00489-f003:**
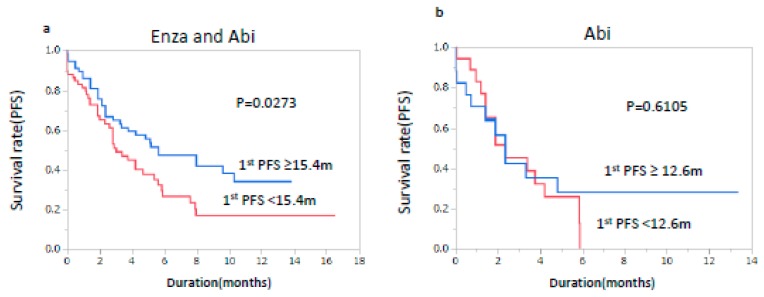
Progression-free survival according to the progression-free survival of first-line therapy. (**a**) Progression-free survival according to a median progression-free survival of first-line therapy <15.4 months or ≥15.4 months as the cut-off values in enzalutamide- and abiraterone-treated patients. (m, months). (**b**) Progression-free survival according to the progression-free survival of first-line therapy <12.0 months or ≥12.0 months as the cut-off values in enzalutamide-treated patients. (**c**) Progression-free survival according to a median progression-free survival of first-line therapy <17.9 months or ≥17.9 months as the cut-off values in enzalutamide-treated patients. (**d**) Progression-free survival according to a median progression-free survival of first-line therapy <12.6 months or ≥12.6 months as the cut-off values in abiraterone-treated patients.

**Table 1 jcm-08-00489-t001:** Patient’s backgrounds.

	Value	Range/%
Enza as initial therapy	82	
Abi as initial therapy	25	Total 107
Enza as second-line therapy	7	
Abi as second-line therapy	21	Total 28
Median age	73.0	54-88
Median BMI (kg/m^2^)	23.4	16.09-34.06
Median TST at biopsy (ng/dL)	457.5	228-847
Median PSA at biopsy (ng/mL)	79.5	3.43-15332
Median PSA at Enza/Abi/total (ng/mL)	30.1/41.1/34.1	0.59–5942.62/3.52–13296/0.59–13296
Median TST at Enza/Abi/total (ng/dL)	13/12/13	2–92/3–31/2–92
Median previous treatment course number	3	1 to 5
Median EOD score 0/1/2/3/4	2	7/23/18/28/15
Median follow-up period (month)	68.3	11.81–241.60
Median Enza/Abi PFS period (month)	3.9/2.1	0–16.50/0–13.37
Median first-line PFS (month)	15.9	0.50–171.40
Gleason Score sum (N)		
≤6	4	4.17%
7	15	15.63%
8	25	26.04%
≥9	52	54.17%
Bone mets	89	83.18%
Lymph mets	34	31.78%
Lung mets	9	8.41%
Liver mets	6	5.61%
No mets	12	11.21%
Patients who died	23	21.50%
Pre-/post-docetaxel	43/64	40.19/59.81%
Steroid use	43	40.19%
Estramustine use	50	46.73%
Enzalutamide dose	160 mg/80 mg	84/5
Abiraterone dose	1000 mg/750 mg	45/1

Enza: enzalutamide; Abi: abiraterone; BMI: body mass index; PFS: progression-free survival; Mets: metastasis; PSA: prostate-specific antigen; TST: target serum testosterone; EOD: extent of disease.

**Table 2 jcm-08-00489-t002:** Predictive factors of PFS in enzalutamide- and abiraterone- treated patients.

	Univariate Analysis				Multivariate Analysis		
	Cut off	HR	COI	*P*	HR	COI	*P*
Age	72	0.75	0.48–1.16	0.1932			
GS	9	0.93	0.59–1.48	0.7654			
First-line PFS (m)	15.4	0.60	0.37–0.95	0.0309	0.85	0.215–3.25	0.8126
Previous docetaxel	+/−	2.44	1.575–3.81	<0.0001	1.50	0.43–5.36	0.5222
Liver mets	+/−	1.69	0.70–3.46	0.2186			
Visceral mets	+/−	1.51	0.83–2.58	0.1682			
Lymph mets	0	1.67	1.07–2.63	0.0249	2.32	0.71–7.90	0.1607
High volume	+/−	1.73	1.09–2.83	0.0198	2.47	0.45–13.74	0.2927
EOD score	2	2.79	1.52–5.47	0.0004	2.79	0.65–14.57	0.1721
ALP (ng/dL)	254	1.20	0.78–1.86	0.4057			
ICTP (ng/mL)	6.6	1.40	0.71–2.78	0.3307			
Hb (g/dL)	11.9	0.65	0.42–1.01	0.0538			
LDH (mg/dL)	212	0.98	0.64–1.52	0.9446			
Alb (g/dL)	3.9	0.85	0.54–1.35	0.5001			
CRP (mg/dL)	0.15	1.87	1.11–3.30	0.0173	0.67	0.17–3.10	0.5871
NLR (ng/dL)	2.6	1.13	0.66–1.94	0.6593			
PSA (ng/mL)	34.1	1.73	1.12–2.71	0.0143	1.24	0.44–3.66	0.6899
TST (ng/dL)	13	0.26	0.13–0.51	<0.0001	0.31	0.10–0.93	0.0365 *
Steroid use	+/−	2.24	1.44–3.54	0.0003	1.10	0.40–2.95	0.8484
Estramustine use	+/−	1.69	1.09–2.65	0.0179	1.41	0.46–4.39	0.5442

Pre-treatment course: previous treatment course; GS: Gleason Score; ALP: alkaline phosphatase; ICTP: I collagen telopeptide; Hb: hemoglobin; LDH: Lactate dehydrogenase; Alb: albumin; CRP: C-reactive protein; NLR: Neutrophil/Lymphocyte ratio; PSA: prostate-specific antigen; * Statistical significance *p* < 0.05.

**Table 3 jcm-08-00489-t003:** Comparison of clinical factors in patients with TST < 13 ng/dL and TST ≥ 13 ng/dL.

	TST < 13	TST ≥ 13	*P*-Value
	Median (Average)	Median (Average)	
Age	71.50(68.74)	70.00(70.88)	0.2298 †
1st-line PFS (month)	29.40(36.33)	14.63(32.755)	0.3576 ††
Pre-docetaxel	42.11%(16/38)	23.53%(8/34)	0.1336 †††
GS≥9	41.94%(13/31)	50%(16/30)	0.6159 †††
Lymph mets	54.29%(19/35)	38.24%(13/34)	0.2301 †††
Bone mets	92.11%(35/38)	88.24%(30/34)	0.7002 †††
Liver mets	5.56%(2/36)	14.71%(5/34)	0.2533 †††
Visceral mets	8.33%(3/36)	23.53%(8/34)	0.1062 †††
High volume	65.71%(23/35)	50.00%(17/34)	0.2270 †††
EOD score	2(2.24)	2(1.97)	0.3965 †
BSI	0.48(2.16)	0.59(2.33)	0.9342 ††
ALP (ng/dL)	209(647.87)	263.50(603.50)	0.0158 *,††
PSA (ng/mL)	33.34(710.73)	39.66(110.72)	0.8391 ††
CRP (mg/dL)	0.42(1.09)	0.50(1.53)	0.7866 ††
PSA at biopsy (ng/mL)	58.57(473.01)	72.95(563.79)	0.3903 ††
TST at biopsy (ng/dL)	4.42(4.22)	3.41(3.32)	0.1544 †
TST nadir at 1st line (ng/dL)	9(38.69)	12(13.23)	0.8974 ††

BSI: bone scan index; * Statistical significancer *p* < 0.05; † welch, †† wilcoxon, ††† fisher.

**Table 4 jcm-08-00489-t004:** Predictive factors of overall survival (OS) in enzalutamide- and abiraterone-treated patients.

	Univariate Analysis				Multivariate Analysis		
	Cut off	HR	COI	*P*	HR	COI	*P*
Age	72	0.94	0.54–1.66	0.8371			
GS	9	0.81	0.45–1.46	0.4801			
First-line PFS (m)	15	0.70	0.36–1.23	0.2134			
Previous docetaxel	+/−	2.38	1.35–4.31	0.0025	3.04	1.41–7.175	0.0038 *
Liver mets	+/−	4.35	1.87–8.92	0.0014	0.35	0.07–1.905	0.2079
EOD score	2	2.87	1.305–7.56	0.0069	1.64	0.56–5.58	0.3778
Visceral mets	+/−	2.90	1.45–5.40	0.0038	7.88	1.62–29.52	0.0139 *
Lymph mets	+/−	2.13	1.21–3.80	0.0091	1.60	0.77–3.29	0.2041
High volume	+/−	2.65	1.38–5.63	0.0028	1.60	0.64–4.50	0.3266
ALP (ng/dL)	254	3.04	1.71–5.66	0.0001	2.51	1.25–5.265	0.0090 *
ICTP (ng/mL)	6.6	2.26	1.87–6.51	0.0941			
Hb (g/dL)	11.9	0.49	0.27–0.85	0.0109	0.87	0.39–1.86	0.7144
LDH (mg/dL)	212	1.92	1.10–3.43	0.0210	2.95	1.43–6.40	0.0033 *
Alb (g/dL)	3.9	0.61	0.33–1.10	0.0990			
CRP (mg/dL)	0.15	1.93	0.97–4.28	0.0607			
NLR (ng/dL)	2.6	1.93	0.93–4.18	0.0790			
PSA (ng/mL)	34.1	3.11	1.72–5.97	0.0001	1.76	0.88–3.69	0.1101
TST (ng/dL)	13	0.99	0.44–2.55	0.8750			
Steroid use	+/−	1.11	0.64–1.95	0.7011			
Estramustine use	+/−	0.95	0.55–1.66	0.8567			

* Statistical significance *p* < 0.05.

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
