# Peer review of "Higher Serum Testosterone Levels Associated with Favorable Prognosis in Enzalutamide- and Abiraterone-Treated Castration-Resistant Prostate Cancer"

_jcm, 2019, doi:10.3390/jcm8040489_

Round 1
Reviewer 1 Report
Substitute "running Head" with "running title"
Introduction section should be improved focusing the attention also on androgen and estrogen receptors that exert a pivotal role in PC initiation and progression. To obtain useful information, authors could read the following manuscripts . doi: 10.3389/fonc.2018.00002, doi: 10.18632/oncotarget.6220.
Tables 1 and 2 are lacking. Did the authors refer to supplementary Tables?
There are no information about the Gleason's Score.
I think that the authors analyzed the Androgen receptors or other proteins that are currently screened, but there are no information.
Authors should emphasize the clinical features of the patients that induced the treatment choice.
"Definition of high-volume tumor" seems a very strange definition. Did You performed any measure of tumor size?
In the list of abbreviations different acronyms are lacking.
Author Response
Reviewer 1
Point 1
Substitute "running Head" with "running title"
Response 1
Thank you very much. We have changed running as suggested.
Point 2
Introduction section should be improved focusing the attention also on androgen and estrogen receptors that exert a pivotal role in PC initiation and progression.
To obtain useful information, authors could read the following manuscripts . doi: 10.3389/fonc.2018.00002, doi: 10.18632/oncotarget.6220.
Response 2
We have reconstructed and cited the indicated Ref in the introduction part.
Point 3
Tables 1 and 2 are lacking. Did the authors refer to supplementary Tables?
Response 3
We are so sorry for the mistake in the upload of tables. We will add the table 1-4.
Point 4
There are no information about the Gleason's Score.
Response 4
We have included Gleason's Score in proportional hazard analysis in Table 1-4.
As shown in table, Gleason's Score did not remain as an independent prognostic factor.
Point 5
I think that the authors analyzed the Androgen receptors or other proteins that are currently screened, but there are no information.
Response 5
The point is really important. We are currently studying the association AR mutation and amplification in cell free DNA and prognosis in patients received novel AR targeted drugs. AR amplification seems to associated with serum testosterone. However, the contents will be included in the next paper. The current manuscripts we would like to focus on the serum testosterone.
Point 6
Authors should emphasize the clinical features of the patients that induced the treatment choice.
Response 6
Thank you very much. We have listed the patients back ground in Table 1. As reviewer suggested, majority of patients received Enza as a fist line. So we have described in the limitation section as follows “Fourth, due to the prior introduction of Enza in the Japanese healthcare system, the majority of Enza group (92%) received the drugs as a 1st line novel AR-targeted drugs, while of a half patients (54%) received Abi as a 1st line AR target drugs. The different result observed between Enza and Abi, including association with serum TST level, may possibly be affected by the background of patients received the drugs. We are currently performing a prospective study to re-assess the role of serum TST in the initial usage of Enza and Abi. The effect of the difference in the clinical features of the patients that induced the treatment choice in the current manuscripts will be answered in the near future.”
Point 7
"Definition of high-volume tumor" seems a very strange definition. Did You performed any measure of tumor size?
Response 7
The size of the tumor was not measured. The number of metastasis and location was counted. The definition of high volume was defined base on the land mark paper of CHAARTED trial [defined as the presence of visceral metastases or ≥4 bone lesions with ≥1 beyond the vertebral bodies and pelvis]. (N Engl J Med 2015; 373:737-746 DOI: 10.1056/NEJMoa1503747)
Point 8
In the list of abbreviations different acronyms are lacking.
Response 8
We have added list of abbreviations as indicated in abbreviations section on page 2.
Reviewer 2 Report
This study investigated the association between total TST and PFS in CRPC patients. It is interesting that TST could be the prognostic factor for relapsed prostate cancer patients. In addition, the difference between ENZ and Abi can be of help for determining the drugs for CRPC patients. However, several points should be improved.
1) I could not find table 1-4 in the manuscript.
2) Introduction is poorly described. The authors should write the background of prostate cancer research, problems in clinical setting and past reports about the diagnostic value of TST measurements.
3) The authors mentioned the diagnostic value of TST before the treatment. To understand the difference of effects between ENZ and Abi, the response of TST or PSA to these treatments should also shown.
3) Ln126: It is quite confusing that TST > 12 is used as a cut-off value. In Ln122, TST > 13 was used.
4) High TST (>13) does not have a prognostic value for overall survival of patients (Fig. 1d). How do the authors explain this result?
5) Discussion: Ln199-208. I could not understand what the authors intended to mean in this paragraph. They should illustrate clearly what is shown by each report they cited.
6) Ln215-218: The reference of Ryan et al is not appropriate. The result of Ryan et al is not consistent with that of this study because Ryan et al showed that high TST is associated with the patients treated with Abi not ENZ. In addition, the discussion in Ln.226- 239 is also contradicted with this report.
7) Overall, total TST has been previously reported to be associated with prognosis of CRPC patients in several studies. The authors should clearly focus on the novel aspect of this study.
Author Response
Reviewer 2
Thank you very much for the valuable comments. We will answer respective points.
1) I could not find table 1-4 in the manuscript.
1) We are so sorry for the mistake in the upload of tables. We will add the table 1-4.
| Table 1. Patient’s backgrounds | ||
| Value | Range / % | |
| Enza as an initial therapy | 82 | |
| Abi as an initial therapy | 25 | Total 107 |
| Enza as a second line therapy | 7 | |
| Abi as a second line therapy | 21 | Total 28 |
| Median Age | 73.0 | 54-88 |
| Median BMI (kg/m2) | 23.4 | 16.09-34.06 |
| Median TST at biopsy (ng/dL) | 457.5 | 228-847 |
| Median PSA at Biopsy (ng/ml) | 79.5 | 3.43-15332 |
| Median PSA at Enza/Abi/total (ng/ml) | 30.1/41.1/34.1 | 0.59-5942.62/3.52-13296/0.59-13296 |
| Median TST at Enza/Abi/total (ng/dL) | 13/12/13 | 2-92/3-31/2-92 |
| Median Previous treatment course number | 3 | 1 to 5 |
| Median EOD Score 0/1/2/3/4 | 2 | 7/23/18/28/15 |
| Median Follow up Period (month) | 68.3 | 11.81-241.60 |
| Median Enza/Abi PFS period (month) | 3.9/2.1 | 0-16.50/0-13.37 |
| Median 1st PFS (month) | 15.9 | 0.50-171.40 |
| Gleason Score sum (N) | ||
| ≤6 | 4 | 4.17% |
| 7 | 15 | 15.63% |
| 8 | 25 | 26.04% |
| ≥9 | 52 | 54.17% |
| Bone Mets | 89 | 83.18% |
| Lymph Mets | 34 | 31.78% |
| Lung Mets | 9 | 8.41% |
| Liver Mets | 6 | 5.61% |
| No Mets | 12 | 11.21% |
| Patients with death | 23 | 21.50% |
| Pre/Post Docetaxel | 43/64 | 40.19/59.81% |
| Steroid Use | 43 | 40.19% |
| Estramustine Use | 50 | 46.73% |
| Enzalutamide Dose | 160mg/80mg | 84/5 |
| Abiraterone Dose | 1000mg/750mg | 45/1 |
| BMI; body mass index, PFS; progression free survival, Mets; metastasis, PSA:prostate-specific antigen , TST; Testosterone | ||
| Table 2. Predictive factors of PFS in Enzalutamide and Abiraterone | |||||||||
| Univariate Analysis | Multivariate Analysis | ||||||||
| Cut off | HR | COI | P | HR | COI | P | |||
| Age | 72 | 0.75 | 0.48 -1.16 | 0.1932 | |||||
| GS | 9 | 0.93 | 0.59-1.48 | 0.7654 | |||||
| 1st line PFS (m) | 15.4 | 0.60 | 0.37-0.95 | 0.0309 | 0.85 | 0.215-3.25 | 0.8126 | ||
| Previous Docetaxel | +/- | 2.44 | 1.575 -3.81 | <.0001< td=""> | 1.50 | 0.43-5.36 | 0.5222 | ||
| Liver Mets | +/- | 1.69 | 0.70-3.46 | 0.2186 | |||||
| Visceral Mets | +/- | 1.51 | 0.83-2.58 | 0.1682 | |||||
| Lymph Mets | 0 | 1.67 | 1.07 -2.63 | 0.0249 | 2.32 | 0.71-7.90 | 0.1607 | ||
| High Volume | +/- | 1.73 | 1.09-2.83 | 0.0198 | 2.47 | 0.45-13.74 | 0.2927 | ||
| EOD Score | 2 | 2.79 | 1.52-5.47 | 0.0004 | 2.79 | 0.65-14.57 | 0.1721 | ||
| ALP (ng/dL) | 254 | 1.20 | 0.78-1.86 | 0.4057 | |||||
| ICTP (ng/ml) | 6.6 | 1.40 | 0.71-2.78 | 0.3307 | |||||
| Hb (g/dL) | 11.9 | 0.65 | 0.42-1.01 | 0.0538 | |||||
| LDH (mg/dL) | 212 | 0.98 | 0.64 -1.52 | 0.9446 | |||||
| Alb (g/dL) | 3.9 | 0.85 | 0.54 -1.35 | 0.5001 | |||||
| CRP (mg/dL) | 0.15 | 1.87 | 1.11 -3.30 | 0.0173 | 0.67 | 0.17-3.10 | 0.5871 | ||
| NLR (ng/dL) | 2.6 | 1.13 | 0.66-1.94 | 0.6593 | |||||
| PSA (ng/ml) | 34.1 | 1.73 | 1.12 -2.71 | 0.0143 | 1.24 | 0.44-3.66 | 0.6899 | ||
| TST (ng/dL) | 13 | 0.26 | 0.13-0.51 | <.0001< td=""> | 0.31 | 0.10-0.93 | 0.0365 | * | |
| Steroid use | +/- | 2.24 | 1.44 -3.54 | 0.0003 | 1.10 | 0.40-2.95 | 0.8484 | ||
| Estramustine use | +/- | 1.69 | 1.09-2.65 | 0.0179 | 1.41 | 0.46-4.39 | 0.5442 | ||
| BMI; body mass index, Pre-treatment course; previous treatment course, Mets; metastasis, ALP; Alkaline phosphatase, ICTP; I collagen telopeptide, LDH; Lactate Dehydrogenase , Alb; albumin, CRP; C-reactive protein , NLR; Neutrophil / Lymphocyte ratio, PSA:prostate-specific antigen , TST; Testosterone, * Statistical significancer p<0.05< td=""> | |||||||||
| Table 3. Comparison of clinical factors between TST<13ng/dL and TST≥13ng/dL | |||||
| TST< 13 | TST≥13 | ||||
| Median(Average) | Median(Average) | P-value | |||
| Age | 71.50(68.74) | 70.00(70.88) | 0.2298 | † | |
| 1st PFS (month) | 29.40(36.33) | 14.63(32.755) | 0.3576 | †† | |
| Pre-Docetaxel | 42.11%(16/38) | 23.53%(8/34 ) | 0.1336 | ††† | |
| GS≥9 | 41.94%(13/31) | 50%(16/30) | 0.6159 | ††† | |
| Lymph Mets | 54.29%(19/35) | 38.24%(13/34 ) | 0.2301 | ††† | |
| Bone Mets | 92.11% (35/38 ) | 88.24% (30/34 ) | 0.7002 | ††† | |
| Liver Mets | 5.56% (2/36 ) | 14.71% (5/34 ) | 0.2533 | ††† | |
| Visceral Mets | 8.33% (3/36 ) | 23.53% (8/34) | 0.1062 | ††† | |
| High Volume | 65.71% (23/35) | 50.00% (17/34 ) | 0.2270 | ††† | |
| EOD Score | 2(2.24) | 2(1.97) | 0.3965 | † | |
| BSI | 0.48(2.16) | 0.59(2.33) | 0.9342 | †† | |
| ALP (ng/dL) | 209(647.87) | 263.50(603.50) | 0.0158 | * | †† |
| PSA (ng/ml) | 33.34(710.73) | 39.66(110.72) | 0.8391 | †† | |
| CRP (mg/dL) | 0.42(1.09) | 0.50(1.53) | 0.7866 | †† | |
| PSA at biopsy (ng/ml) | 58.57(473.01) | 72.95(563.79) | 0.3903 | †† | |
| TST at biopsy (ng/dL) | 4.42(4.22) | 3.41(3.32) | 0.1544 | † | |
| TST nadir at 1st line (ng/dL) | 9(38.69) | 12(13.23) | 0.8974 | †† | |
| PFS; progression free survival, GS; gleason score, Mets; metastasis, EOD; extent of disease, BSI; bone scan index, ALP; Alkaline phosphatase, ICTP; I collagen telopeptide, LDH; Lactate Dehydrogenase , CRP; C-reactive protein , PSA:prostate-specific antigen , TST; Testosterone, * Statistical significancer p<0.05< td=""> | |||||
| † welch, †† wilcoxon,††† fisher | |||||
| Table 4. Predictive factors of OS in Enzalutamide and Abiraterone | |||||||||
| Univariate Analysis | Multivariate Analysis | ||||||||
| Cut off | HR | COI | P | HR | COI | P | |||
| Age | 72 | 0.94 | 0.54-1.66 | 0.8371 | |||||
| GS | 9 | 0.81 | 0.45-1.46 | 0.4801 | |||||
| 1st line PFS (m) | 15 | 0.70 | 0.36-1.23 | 0.2134 | |||||
| Previous Docetaxel | +/- | 2.38 | 1.35-4.31 | 0.0025 | 3.04 | 1.41-7.175 | 0.0038 | * | |
| Liver Mets | +/- | 4.35 | 1.87-8.92 | 0.0014 | 0.35 | 0.07-1.905 | 0.2079 | ||
| EOD Score | 2 | 2.87 | 1.305-7.56 | 0.0069 | 1.64 | 0.56-5.58 | 0.3778 | ||
| Visceral Mets | +/- | 2.90 | 1.45-5.40 | 0.0038 | 7.88 | 1.62-29.52 | 0.0139 | * | |
| Lymph Mets | +/- | 2.13 | 1.21-3.80 | 0.0091 | 1.60 | 0.77-3.29 | 0.2041 | ||
| High Volume | +/- | 2.65 | 1.38-5.63 | 0.0028 | 1.60 | 0.64-4.50 | 0.3266 | ||
| ALP (ng/dL) | 254 | 3.04 | 1.71-5.66 | 0.0001 | 2.51 | 1.25-5.265 | 0.0090 | * | |
| ICTP (ng/ml) | 6.6 | 2.26 | 1.87-6.51 | 0.0941 | |||||
| Hb (g/dL) | 11.9 | 0.49 | 0.27-0.85 | 0.0109 | 0.87 | 0.39-1.86 | 0.7144 | ||
| LDH (mg/dL) | 212 | 1.92 | 1.10-3.43 | 0.0210 | 2.95 | 1.43-6.40 | 0.0033 | * | |
| Alb (g/dL) | 3.9 | 0.61 | 0.33-1.10 | 0.0990 | |||||
| CRP (mg/dL) | 0.15 | 1.93 | 0.97-4.28 | 0.0607 | |||||
| NLR (ng/dL) | 2.6 | 1.93 | 0.93-4.18 | 0.0790 | |||||
| PSA (ng/ml) | 34.1 | 3.11 | 1.72-5.97 | 0.0001 | 1.76 | 0.88-3.69 | 0.1101 | ||
| TST (ng/dL) | 13 | 0.99 | 0.44-2.55 | 0.8750 | |||||
| Steroid use | +/- | 1.11 | 0.64-1.95 | 0.7011 | |||||
| Estramustine use | +/- | 0.95 | 0.55-1.66 | 0.8567 | |||||
| BMI; body mass index, Pre-treatment course; previous treatment course, Mets; metastasis, ALP; Alkaline phosphatase, ICTP; I collagen telopeptide, LDH; Lactate Dehydrogenase , Alb; albumin, CRP; C-reactive protein , NLR; Neutrophil / Lymphocyte ratio, PSA:prostate-specific antigen , TST; Testosterone, * Statistical significancer p<0.05< td=""> | |||||||||
2) Introduction is poorly described. The authors should write the background of prostate cancer research, problems in clinical setting and past reports about the diagnostic value of TST measurements.
2) Thank you very much. We have reconstructed the introduction as shown in red colors.
3) The authors mentioned the diagnostic value of TST before the treatment. To understand the difference of effects between ENZ and Abi, the response of TST or PSA to these treatments should also shown.
3) Thank you very much for the comments. This is an important point. We have added the supplementary table 5 showing the PSA response rate in the sequential usage of Enza or Abi.
| Supplementary Table 5. PSA response in the sequential use of novel AR targeted drugs. | |||||||
| All patients | 1st Abi | Enza→Abi | p-Value | 1st Enz | Abi→Enza | p-Value | |
| 30%PSA response | |||||||
| % (median) | 42.70% | 42.86% | 26.32% | 0.1589 | 42.62% | 28.57% | 0.3331 |
| 50%PSA response | |||||||
| % (median) | 37.10% | 35.71% | 21.05% | 0.1862 | 37.70% | 28.57% | 0.5208 |
| PFS | |||||||
| week (median) | 12.4 | 15.7 | 9.93 | 0.0764 | 12 | 11.1 | 0.2726 |
| 95% CI | 15.5-20.9 | 12.2-24.3 | 8.47-16.0 | 15.1-21.3 | 7.5-19.3 | ||
| PFS; prgression free survival, Enza; Enzalutamide, Abi; Abiraterone, 1st Abi/Enza; Abi/Enz as 1st line therapy, Enz→Abi; Abi use after Enz, Abi→Enza; Enza use after Abi. | |||||||
In terms of serum testosterone, due to the regulation of national healthcare system, we could not follow the value of TST after the treatment with Enza or Abi. However, in the limited patients, Enza usage caused increased TST value (10-20%), while in Abi, TST reduced to the undetectable value.
4) Ln126: It is quite confusing that TST > 12 is used as a cut-off value. In Ln122, TST > 13 was used.
4) Thank you very much. I know this may be confusing. The median TST at the initiation of Enza was 13 ng/dL, and the median TST at the initiation of Abi was 12 ng/dL. The median of Enza and Abi overall was 13ng/dL. So, we have used TST 13ng/dL and 12ng/dL. In order to be objective, to avoid any subjective intention, we have used median value throughout the study in all the clinical factors.
5) High TST (>13) does not have a prognostic value for overall survival of patients (Fig. 1d). How do the authors explain this result?
5) This is a very important point. There are several reasons that we speculate. 1st, as shown in table 3, TST>13ng/dL group showed the significantly higher ALP and relatively higher liver mets, GS>9. Although there seems to be an advantage in the PFS, high ALP, liver mets and GS>9 components may resulted in the limited response to the following treatments such as docetaxel. 2nd, median PFS of Abi or Enza was 3-4 month, that limited the advantage of PFS on OS.
6) Discussion: Ln199-208. I could not understand what the authors intended to mean in this paragraph. They should illustrate clearly what is shown by each report they cited.
6) Thank you very much. We have reconstructed the discussion as shown in red color on page 7. What we try to describe was that, even at lower TST level<20ng high="" serum="" tst="">13ng/dL may represent the remaining AR dependency that can only be targeted by novel AR targeted drugs.
7) Ln215-218: The reference of Ryan et al is not appropriate. The result of Ryan et al is not consistent with that of this study because Ryan et al showed that high TST is associated with the patients treated with Abi not ENZ.
7) Thank you very much. We have reconstructed the discussion as follows on page 7.
The present data correlated with the findings of The clinical advantage of high serum TST was also reported by Ryan et al.10 Although the TST cut-off value was even lower (>8.6 ng/dL) compared to the present study, they found a higher serum TST to be related to a better response to Abi. On the other hand, in our study, the clinical advantage of high serum TST was not found in Abi but found in Enza.
8) In addition, the discussion in Ln.226- 239 is also contradicted with this report.
8) Thank you very much. We found long 1st line PFS was related to the favorable PFS in Enza but not in Abi. We think the result was similar to the finding of the Loriot et al., in Eur J Cancer 2015, 51 (14), 1946-52 that indicated TTCRPC>12 month was related to the good PFS in Enza.
9) Overall, total TST has been previously reported to be associated with prognosis of CRPC patients in several studies. The authors should clearly focus on the novel aspect of this study.
9) Thank you very much. We totally agree. We have changed the conclusion part as follows.
Conclusion
A higher serum TST level (≥13 ng/dL) was associated with a favorable prognosis in Enza and/or Abi-treated patients. Serum TST level 13ng/dL may represent the sign of favorable response to novel AR targeted drugs in CRPC patients.
Round 2
Reviewer 1 Report
Authors improved the quality of the manuscript according to reviewers suggestions.
Author Response
Thank you very much for help us improve the manuscript.
Reviewer 2 Report
The authors responded to my concerns.
In Introduction, the authors added a description about estrogen receptor (ER) function in CRPC. It is interesting that female hormone is also important in CRPC treatment. However, I can not understood how this description is associated with this study. I wonder if the authors expect TST is affected by estrogen signals. I could not find other description regarding the selection of ER-blocker in clinical data the authors presented. How did the atuhros select ER-blokcker, AR-blocker (Abi and Enz) or combination of both?
Author Response
Thank you so much for the bring up the very important points
We will try to answere the question from the reviewer.
Question
It is interesting that female hormone is also important in CRPC treatment. However, I can not understood how this description is associated with this study. I wonder if the authors expect TST is affected by estrogen signals. I could not find other description regarding the selection of ER-blocker in clinical data the authors presented. How did the atuhros select ER-blokcker, AR-blocker (Abi and Enz) or combination of both?
Answer
We at first treat with androgen signal bocker, however, after became refractory to the AR targeted drugs, we swich to the ER stimulator such as Estramustine. It actually work in some goup of patients. That is why 46.7% of patients previously received Estramustine in table 1.
As listed in table 2, Estramustine use was related to resistant to the Enza and Abi response as indicated as HR 1.69 P=0.0179.
I do not know exact mechanism, however, there seems to be cross talk between ER and AR signaling.